# The Transcription Factor Twist1 Has a Significant Role in Mycosis Fungoides (MF) Cell Biology: An RNA Sequencing Study of 40 MF Cases

**DOI:** 10.3390/cancers15051527

**Published:** 2023-02-28

**Authors:** Marjaana J. Häyrinen, Jenni Kiiskilä, Annamari Ranki, Liisa Väkevä, Henry J. Barton, Milla E. L. Kuusisto, Katja Porvari, Hanne Kuitunen, Kirsi-Maria Haapasaari, Hanna-Riikka Teppo, Outi Kuittinen

**Affiliations:** 1Institute of Clinical Medicine, Faculty of Health Medicine, University of Eastern Finland, 70210 Kuopio, Finland; 2Cancer Research and Translational Medicine Research Unit, University of Oulu, 90014 Oulu, Finland; 3Department of Skin and Allergic Diseases, University of Helsinki, Helsinki University Central Hospital, P.O. Box 160, 00029 HUS Helsinki, Finland; 4Genevia Technologies Oy, 33100 Tampere, Finland; 5Department of Haematology, Oulu University Hospital, 90220 Oulu, Finland; 6Medical Research Center Oulu, Oulu University Hospital, University of Oulu, 90220 Oulu, Finland; 7Cancer Center, Oulu University Hospital, 90220 Oulu, Finland; 8Department of Pathology, Oulu University Hospital, 90220 Oulu, Finland; 9Cancer Center, Kuopio University Hospital, 70210 Kuopio, Finland

**Keywords:** cutaneous T-cell lymphoma, mycosis fungoides, Twist1, Zeb1, RNA sequencing, DNA methylation, laser capture microdissection

## Abstract

**Simple Summary:**

Mycosis fungoides (MF) is the most common variety of cutaneous T-cell lymphoma. Our previous studies showed that the epithelial–mesenchymal transition (EMT) transcription factors (TFs) Twist1 and Zeb1 have prognostic value in MF. The main objective of the present study was to gain better knowledge about the biological mechanisms behind this phenomenon. The RNA of 40 skin tumor biopsies (from 40 patients) was sequenced and analyzed. Twist1 protein expression seemed to classify MF cases into different groups based on their global RNA expression. Additionally, high Twist1 protein expression was associated with several genes and pathways known to have roles in aggressive tumor biology. For Zeb1, similar results were not found. Our results suggest Twist1 to be a central transcription factor and pathway regulator in the disease progression of MF. Twist1 might be an interesting object for developing targeted therapies for MF.

**Abstract:**

The purpose of this RNA sequencing study was to investigate the biological mechanism underlying how the transcription factors (TFs) Twist1 and Zeb1 influence the prognosis of mycosis fungoides (MF). We used laser-captured microdissection to dissect malignant T-cells obtained from 40 skin biopsies from 40 MF patients with stage I–IV disease. Immunohistochemistry (IHC) was used to determinate the protein expression levels of Twist1 and Zeb1. Based on RNA sequencing, principal component analysis (PCA), differential expression (DE) analysis, ingenuity pathway analysis (IPA), and hub gene analysis were performed between the high and low Twist1 IHC expression cases. The DNA from 28 samples was used to analyze the TWIST1 promoter methylation level. In the PCA, Twist1 IHC expression seemed to classify cases into different groups. The DE analysis yielded 321 significant genes. In the IPA, 228 significant upstream regulators and 177 significant master regulators/causal networks were identified. In the hub gene analysis, 28 hub genes were found. The methylation level of TWIST1 promoter regions did not correlate with Twist1 protein expression. Zeb1 protein expression did not show any major correlation with global RNA expression in the PCA. Many of the observed genes and pathways associated with high Twist1 expression are known to be involved in immunoregulation, lymphocyte differentiation, and aggressive tumor biology. In conclusion, Twist1 might be an important regulator in the disease progression of MF.

## 1. Introduction

Primary cutaneous T-cell lymphomas (CTCLs) are a heterogeneous group of non-Hodgkin’s lymphomas with no evidence of extracutaneous spread at the time of diagnosis. Mycosis fungoides (MF) is the most common CTCL, accounting for about 60% of all CTCL cases [1]. Usually, patients are adults, and most of the patients are males. Sézary syndrome (SS) is a leukemic variant of CTCL.

The clinical course of MF is variable. The disease starts with localized or disseminated patches or plaques that can remain skin-limited for years. However, in a subset of patients (10–20%) [2] the disease evolves into the tumor or erythroderma stages, including extracutaneous spread with poor prognosis. Although the treatment modalities for MF have developed over the years, the disease remains still incurable, indicating the need to understand the biology of this disease better.

Epithelial-to-mesenchymal transition (EMT) is an essential process in embryonic development and common in cancer progression [3,4,5]. During EMT, the epithelial cells obtain the mesenchymal phenotype and convert to a more invasive, motile phenotype and acquire resistance to apoptosis. The role of EMT in cancer progression, dissemination, and therapy resistance has been well recognized in epithelial tumors, but in the case of hematopoietic malignancies, the significance of EMT is less well studied. One basic difference is that the cells of hematological malignancies already have a mesenchymal phenotype since they arrive from blood cells derived from the embryonic mesoderm. However, some of the EMT-controlling transcription factors (EMT–TFs), including Twist1 and Zeb1, control the differentiation of hematopoietic cells and have been associated with the progression of hematological malignancies [6]. Twist1 is a T-cell oncoprotein that belongs to the basic helix–loop–helix (bHLH) protein family. Twist1 regulates the inflammatory processes and is involved in lymphocyte function and maturation, working as a key regulator of immune cells, especially T helper (Th) cell activation [7,8]. Zeb1 is a protein-coding gene that suppresses hematopoiesis and downregulates the expression of CD4 during T-cell maturation [9]. In the study of vanDoorn et al. (2004) [10], Twist1 was highly overexpressed among SS patients. Goswami et al. (2012) [11] showed that Twist1 expression was correlated with MF and SS stages. They also observed an association between increased Twist1 and c-Myc expression and abnormal p53 expression.

Our earlier studies showed that IHC Twist1+ is associated with worse prognosis and Zeb1+ with better prognosis in patients with MF [12]. In this study, our aim was to investigate both the up- and downstream regulation of Twist1 and Zeb1 to understand the biology behind their prognostic value. Our methods included Twist1 and Zeb1 IHC and RNA sequencing from 40 MF cases. The methylation level of the Twist1 promoter was analyzed from 28 MF samples.

## 2. Materials and Methods

### 2.1. Patient Material

The retrospective patient material consisted of 40 biopsies from 40 MF patients with stage I–IV disease from the Helsinki University Hospital obtained during the years 2015–2019. 21 formalin-fixed and paraffin-embedded (FFPE) samples were taken at the time of the diagnosis and 19 from patients with relapsed disease.

Patient data were collected from hospital records. These clinical variables included gender, age, WHO-EORTC stage, plasma lactate dehydrogenase (LDH) level, treatments, data on follow-up or relapses, progression, and mortality.

### 2.2. Immunohistochemical Staining, Analysis, and Correlation with Disease Presentation and Outcome

Twist1 and Zeb1 immunohistochemical staining was performed as previously described in Lemma et al. (2013) [13]. In the IHC analysis, the cut-off point for low and high expression of Twist1 was 17.6% and that for Zeb1 was 37%, defined by using a receiver operating characteristic (ROC) curve. Morphologically assessed neoplastic cells were counted as positive in the IHC analysis.

Tumor cell count was estimated from the hematoxylin–eosin-stained samples identifying the hyperchromatic small to medium-sized haloed lymphocytes with hyperconvoluted nuclei as a percentage of the surrounding reactive lymphocytic cell infiltrate. The tumor cell count was estimated by an experienced hematopathologist. Additionally, the proportion of the entire lymphocytic infiltrate was estimated from the sample area.

The time from diagnosis to the initiation of systemic therapy or to the last follow-up date (TTST) was calculated and used to perform a Kaplan–Meier analysis. A chi-squared test was used as a statistical method, and statistical significance was evaluated with the log rank. *p*-values < 0.05 were considered statistically significant. IBM SPSS Statistics for Macintosh, Version 28.0. (Armonk, NY: IBM Corp.) was used for ROC curves and Kaplan–Meier analysis.

### 2.3. Microdissection and RNA Extraction

In total, 8–13 sequential paraffin-embedded slide sections with a thickness of 5 µm were prepared and mounted on pet (polyethylene terephthalate) slides (Leica AS LMD; Leica Microsystems Ltd., Wetzlar, Germany). Paraffin was removed by soaking in xylene twice for 10 min, and sections were stained with hematoxylin. One section was stained with CD3 antibody (NCL-L-CD3-565), which was used as a guide to differentiate between lymphocytic and epithelial cells (Appendix A). Laser capture microdissection (LCM) was performed using a ZEISS PALM MicroBeam Microsdissection system (Carl Zeiss Microscopy GmbH, Oberkochen, Germany) and adhesive collection caps.

After microdissection, the samples were placed into collection vessels containing appropriate volumes of Depaffinization Solution (#19093, Qiagen GmbH, Hilden, Germany), and RNA was extracted using an MiRNeasy FFPE extraction kit (#217504, Qiagen GmbH, Hilden, Germany) generally according to the manufacturer’s instructions. However, after adding proteinase K solution, the samples were incubated overnight at 56 °C with gentle shaking for a better yield.

### 2.4. The TWIST1 Promoter Methylation Analysis

After LCM, DNA was isolated using a QIAamp DNA FFPE tissue kit (Qiagen, Hilden, Germany). DNA concentrations were measured using a Qubit 4 fluorometer, and bisulfite treatment was carried out using an EpiTect Fast DNA Bisulfite kit (Cat. No. 59824, Qiagen). The promoter area of the *TWIST1* gene (Entrez Gene ID: 7291) was then amplified using a PyroMark CpG Assay (GeneGlobe Cat. no: PM00030121), PyroMark PCR kit, and RotorGeneQ device (Qiagen). The specificity of biotin-labeled amplification products was confirmed on agarose gel and purified for pyrosequencing with Streptavidin Sepharose beads (Cytiva) using a PyroMark Q24 Vacuum Workstation. Single-stranded DNA on a sequencing plate was annealed with the sequencing primer at 80 ºC (2 min) and cooled at room temperature (15 min). Then, the plate was processed using the PyroMark Q24 Instrument with compatible Gold Q24 reagents. Finally, the sequencing run was analyzed via the PyroMark Q24 software version 2.0.6. (Qiagen, Hilden, Germany).

### 2.5. RNA Sequencing Data Analysis

The quality and quantity of the extracted RNA samples were analyzed with a LabChip GX Touch HT RNA Assay Reagent Kit (PerkinElmer, Waltham, MA, USA) and Qubit RNA BR kit (Thermo Fisher Scientific, Waltham, MA, USA). For genomic DNA contamination measurement, a Qubit DNA BR kit (Thermo Fisher Scientific, Waltham, MA, USA) was used. Dual-indexed mRNA libraries were prepared from 150 ng of total RNA with a QuantSeq 3′ mRNA–Seq Library Prep Kit FWD (Lexogen Gmbh, Vienna, Austria) according to user guide version 015UG009V0251. During second strand synthesis, 6 bp Unique Molecular Identifiers (UMIs) were introduced with the UMI Second Strand Synthesis Module (Lexogen Gmbh, Vienna, Austria) for detection and removal of PCR duplicates. The quality of the libraries was measured with a LabChip GX Touch HT DNA High Sensitivity Reagent Kit (PerkinElmer, Waltham, MA, USA). Sequencing was performed with a NovaSeq 6000 System (Illumina, San Diego, CA, USA) with read length of 2 × 101 bp and target coverage of 10 M reads for each library. QuantSeq 3′ mRNA–Seq Integrated Data Analysis Pipeline version 2.3.1 FWD UMI (Lexogen Gmbh, Vienna, Austria) on Bluebee^®^ Genomics Platform was used for primary quality evaluation of the RNA sequencing data.

### 2.6. Read Counts and Principal Component Analysis

For visual exploration of the data, the read counts were normalized using the variance stabilizing transformation (VST) method implemented in the DESeq2 (version 1.30.1) package [14] in R (version 4.0.3) [15], which transforms the count data in a way that minimizes differences between samples for rows with small counts and normalizes the data with respect to library size, with large values approximating a log2 scale. Visual inspection of the samples was performed using principal component analysis (PCA) implemented in the ‘prcomp’ function in R, applied to the normalized read counts of the top 500 genes according to variance. Additionally, the same data were used to generate a Pearson’s correlation heat map from all pairwise comparisons of samples, using the ‘pheatmap’ package (version 1.0.12) [16] in R.

### 2.7. Differential Expression Analyses

Data normalization and differential expression (DE) analysis were performed using the DESeq2 package [14] in R. Genes with an absolute log2 fold change > 0.58 (absolute fold change of 1.5) and an adjusted *p* value < 0.05 (adjusted for multiple testing using the Benjamini–Hochberg procedure [17]) were considered to be significantly differentially expressed.

Initially, two contrasts were made using only diagnostic samples: first, high-Twist1-expression (Twist+) samples against low-Twist1-expression (Twist1-) samples, followed by high-Zeb1-expression (Zeb1+) samples against low-Zeb1-expression (Zeb1-) samples. A second round of DE analysis was performed between Twist1+ and Twist1- samples, this time including both diagnostic and follow-up samples. The results of the two sets of Twist1 expression contrasts were compared using Pearson’s correlations.

### 2.8. Ingenuity Pathway Analysis

The full results table from the DESeq2 analysis of Twist1+ versus Twist1-, using both diagnostic and follow-up samples, was read into the Ingenuity Pathway Analysis (IPA) software [18]. Then an IPA core analysis was run with the following analysis settings: the reference was set to ‘User Dataset’, the confidence level was set to ‘Experimentally Observed’, the species was set to ‘Human’, the log2 fold-change filter was set to <−0.58 and >0.58, and the adjusted *p* value filter was set to <0.05. The significance threshold for the identified pathways and regulators was also set to an adjusted *p* value of 0.05.

### 2.9. Hub Gene Analysis

To identify hub genes, a protein–protein interaction (PPI) network was constructed from the genes that were significantly differentially expressed between high and low Twist1 expression groups (all samples) using the STRING database [19] through the StringApp [20] within Cytoscape [21]. The connectivity of the nodes in the network was assessed using the cytoHubba plugin [22] and nodes with a degree of connectivity of 10 or more were said to be hub genes.

## 3. Results

### 3.1. Patients

Patient demographics are presented in Table 1. The median age was 63 years (range 19–86 years), and most of the patients were male (70%). The median follow-up time was 32.2 months (range: 6.28–203 months).

### 3.2. Immunohistochemistry of Twist1 and Zeb1, Correlation with Histomorphology, Disease Presentation and Outcome

For Twist1, there were 20 high expression and 20 low expression cases. For Zeb1, there were 4 high expression and 36 low expression cases. Twist1 expression was not associated with tumor cell percentage or lymphocyte cell proportion, or with the clinical stage.

There were no significant correlations between the collected clinical variables and Twist1 protein expression. However, among the patients with diagnostic samples, there was a trend for the cases with high Twist1 protein expression to require systemic therapy sooner than the cases with low expression (*p* value = 0.133, Figure 1).

### 3.3. The Analysis of TWIST1 Promoter Methylation

The methylation levels of four CpG islands of 28 cases were analyzed: CpG1, CpG2, CpG3, and CpG4. The means and standard deviations for GpG islands 1, 2, 3, and 4 were M_1_ = 3.96 (SD_1_ = 2.25), M_2_ = 3.36 (SD_2_ = 2.22), M_3_ = 12.31 (SD_3_ = 3.76), and M_4_ = 1.58 (SD_4_ = 0.902), respectively. For the total methylation level, the mean was 20.95 and SD was 6.88. The methylation levels did not correlate with the IHC or RNA expression of Twist1.

### 3.4. The Association between Twist1 and Zeb1 Protein Levels and RNA Levels

To see how well the RNA expression for Twist1 and Zeb1 corresponded to high/low classification based on IHC expression, the normalized expression for each gene in each sample was plotted on a heat map (Figure 2). Generally, Twist1 RNA expression agreed with the high/low IHC expression classification of the samples, while Zeb1 RNA expression did not correlate well with the classification. A Pearson correlation coefficient was conducted to examine these correlations. Twist1 RNA expression correlated positively to IHC expression (r (38) = 0.46, *p* value < 0.01), while no correlation could be seen between Zeb1 RNA and protein expression (r (38) = −0.070, *p* value = 0.67). For Twist1, there were also cases that did not correlate; for example, the case of MH37 had high Twist1 protein expression but the lowest RNA expression of the whole series.

### 3.5. The Principal Component Analysis (PCA)

To investigate whether the samples clustered according to their Twist1 and Zeb1 expression, we performed PCAs on the normalized read counts both for all samples combined and diagnostic samples individually (see Methods).

The samples were observed to separate along PC1 according to their Twist1 expression category in both analyses. PC1 explained 26% of the variation for all samples (Figure 3a) and 32% of the variation for diagnostic samples (Figure 3b). A similar pattern was not seen for Zeb1 expression categories. Furthermore, there appeared to be no separation between the diagnostic and follow-up samples (Appendix A).

### 3.6. High vs. Low Twist1 and Zeb1—Differential Expression Analysis

Differential expression (DE) analysis between Twist1+ and Twist1- diagnostic samples only returned 11 significantly (adjusted *p* value <= 0.05, absolute log2 fold change > = 0.58) differentially expressed genes: *OAS2*, *ENSG00000201329*, *FCER1G*, *LGALS9*, *LYZ*, *LITAF*, *HLA-DRA*, *HLA-A*, *IGHM*, *NDUFA4* and *RPGR (*Figure 4). The corresponding analysis for Zeb1 expression yielded no significant genes.

Considering the low statistical power when analyzing only diagnostic samples and given that there seemed to be little separation between diagnostic and follow-up samples in terms of gene expression (Appendix A), the DE analysis between Twist1+ and Twist1- samples was repeated for the diagnostic and follow-up samples combined.

This analysis yielded 321 significant genes (adjusted *p* value < = 0.05, absolute log2 fold change > = 0.58). Expression of the top 100 significant genes (according to the adjusted *p* value) is visualized in the heatmap in Figure 5, where the samples are clustered largely by their Twist1 categories. The top genes generally have higher expression in the Twist1+ samples than in the Twist1- samples. Furthermore, diagnostic and follow-up samples appeared to be intermixed, irrespective of the clinical staging.

The results of the DE analysis with all samples combined correlated well with those from the analysis using only diagnostic samples for both the adjusted *p* value (Pearson’s r = 0.63, *p* << 0.01, Figure 6a) and the log2 fold change (Pearson’s r = 0.85, *p* << 0.01, Figure 6b). Seven of the significant genes from the first analysis were also identified in the second analysis (*OAS2*, *ENSG00000201329*, *LGALS9*, *LITAF*, *HLA-DRA*, *IGHM* and *NDUFA4*).

### 3.7. High vs. Low Twist1—Ingenuity Pathway Analysis

The results of the differential expression analysis were used as input for Ingenuity Pathway Analysis (IPA). This resulted in three significant pathways (adjusted *p* value < 0.05): the ‘GP6 Signaling Pathway’, ‘Hepatic Fibrosis/Hepatic Stellate Cell Activation’, and ‘B Cell Development’. Prior to correcting for multiple testing, there were 35 significant pathways (*p* value < 0.05), as shown in Appendix A. The IPA core analysis also identified 228 significant upstream regulators (adjusted *p* value < 0.05) and 177 significant master regulators/causal networks (adjusted *p* value < 0.05) (Appendix A).

### 3.8. Hub Gene Analysis

Analysis of the protein–protein interaction network constructed from the results of the Twist1 differential expression analysis resulted in 28 identified hub genes (connectivity degree >= 10), as shown in Figure 7.

## 4. Discussion

In previous studies, the protein expression of the EMT TFs Twist1 and Zeb1 was shown to have prognostic relevance in MF [12]. Here, we sought to explore the biology of these differences more deeply. Through RNA sequencing, we found that most of the regulation of Twist1 expression occurs at the translational level, while no correlations were found between Zeb1 protein and the mRNA level. In the PCA, Twist1 expression was found to classify MF cases into different clusters according to their global RNA expression. Several genes and pathways known to be associated with aggressive tumor biology were found to be overexpressed among high Twist1 cases. For Zeb1, similar associations were not observed.

MF is the most frequent cutaneous T-cell lymphoma, originating from the peripheral epidermotropic T-cells. Despite the many available treatment options, MF is still considered incurable, except for allogenic stem cell transplantation. The intricate molecular mechanisms behind the MF transition from an indolent to a progressive disease are not completely understood. Currently, it is anticipated that alterations in defined signaling networks promote the proliferation, survival, and migration of malignant T-cells, as well as the suppression of their immune regulation, resulting in changes to the tumor microenvironment that enables disease progression [23].

In our previous study [12], we found that the IHC detection of Twist1 and Zeb1 have prognostic value in MF: Twist1+ and Zeb1- identified patients with a worse prognosis. The results of the IHC analysis of the present study were in line with previous results, but statistical significance could not be shown, likely due to the limited sample size. Other studies have also proposed that Twist1 protein may be one of the key regulators in MF progression. Dobos et al. [24] studied the prognostic value of the expression levels of several proteins of peripheral blood leukocytes in MF and SS using the multiomics method. The authors found T-plastin, Twist1, and KIR3DL2 to bear the highest prognostic relevance. On the other hand, Song et al. [25] demonstrated core oncogenic processes behind large cell transformation of MF. These processes included metabolic reprogramming, cellular plasticity, upregulation of myelocytomatosis oncogene (MYC) and E2 promoter binding factor (E2F) activities, and downregulation of major histocompatibility complex 1 (MHC1). One of the key elements in cellular plasticity is the upregulation of Twist1 protein expression through gene amplification [25].

The regulation of Twist1 expression in cancer is a complicated process including modulation at many levels and depending on the cancer type and tissue context [26]. Transcriptionally, Twist1 can be upregulated via multiple signal transduction pathways such as the tumor necrosis factor receptor (TNFR), receptor tyrosine kinase (RTK), frizzled (FZD), tumor growth factor beta (TGFß), NOTCH, and epidermal growth factor receptor (EGFR) pathways [26]. The most important intracellular regulators include mitogen activated protein kinase (MAPK), protein kinase B (Akt), nuclear factor-κB (NF-κB), muscle segment homeobox 2 (MSX2), ß-catenin, Fibulin 5 (FBLN5), mothers against decapentaplegic homolog 2 (Smad), high-mobility group AT-hook 2 (HMGA2), signal transducer and activator of transcription 3 (STAT3), and hypoxia inducible factor-1 (HIF-1α) [26]. MAPK, Akt, and casein kinase (CK2) are important Twist1 phosphorylating kinases that participate in the post-translational regulation of Twist1 [21].

We found an association between Twist1 protein and RNA expression levels. In contrast to a study by Galvan et al. [27], we did not detect a clear connection between promoter methylation and the RNA levels of *TWIST1*/Twist1, indicating that this is likely not a major reason for *TWIST1* overexpression in MF. We also did not observe overexpression of other known positive *TWIST1* regulators, thereby leaving *TWIST1* overexpression largely unexplained. Despite a robust correlation between *TWIST1* mRNA and protein levels, there were also cases with deviant results, indicating that translational/posttranslational regulation also plays a role. For example, beta-transducing repeat containing protein (β-TRCP) was shown to play a role in Twist1 degradation [28]. For Zeb1, the protein and RNA amounts did not correlate with each other, suggesting that most regulation takes place in the posttranscriptional level.

In the PCA, Twist1 expression partly explained the clustering of the MF cases along the first principal component. This result illustrates that Twist1 is an important modulator of MF biology. This implication seems reasonable considering Twist1’s integral role in T-cell differentiation. However, in this study setting, we were only able to demonstrate association between these two factors, which does not always imply causality. For Zeb1, the PCA did not reveal any grouping of samples according to their Zeb1 expression. The small number of cases with high Zeb1 expression might explain why we could not detect a function of Zeb1 in MF.

Differential expression analysis between Twist1+ and Twist1- diagnostic MF samples revealed 11 significantly differentially expressed genes; *OAS2*, *ENSG00000201329*, *FCER1G*, *LGALS9*, *LYZ*, *LITAF*, *HLA-DRA*, *HLA-A*, *IGHM*, *NDUFA4* and *RPGR*. High Twist1 protein expression was associated with overexpression of *RPGR* and *ENSG0000020132.* The rest of the genes were downregulated when Twist1 protein expression was high. Most of these genes are associated with adverse biological features in different malignancies. High *OAS2* expression was shown to associate with better prognosis in breast [29], bladder [30] and colorectal cancer [31]. In acute myeloid leukemia (AML), *OAS2* expression was shown to induce chemoresistance [32]. Galectin-9 was previously shown to correlate with disease severity and decreased CD8 cell infiltration in CTCL [33]. Moreover, anti-Gal-9 therapy selectively expands intratumoral T-cell immunoglobulin and mucin-domain containing-3 positive (TIM-3+) cytotoxic CD8 T-cells, as well as immunosuppressive regulatory cells [34]. Previous studies have confirmed the role of *FCER1G* in several cancers [35]. *FCER1G* takes part in promoting squamous carcinogenesis (SCC) progression [36], and predicts poor prognosis in gliomas and clear cell renal cell carcinomas (RCC) [37,38]. In multiple myeloma, *FCER1G* predicts better prognosis [39,40]. According to the previous research, *LITAF* may be considered as a tumor suppressor [41,42]. In AML, *LITAF* was shown to increase cell apoptosis and differentiation [43]. In colorectal cancer, HLA-A is associated with a favorable prognosis [44]. The downregulation of *NDUFA4* was detected in RCC [45].

The ingenuity pathway analysis of Twist1 overexpression resulted in 35 pathways before correcting for multiple testing. When Twist1 was overexpressed, the glycol protein VI (GP6) signaling pathway was one of the most strongly upregulated pathways. GP6 is part of the immunoglobulin superfamily and is expressed in the platelets and megakaryocytes taking part in their activation. Along with their coagulative functions, platelets have an active role in regulating immune phenomena and in tumor cells immune escape. On the other hand, platelets also induce EMT in tumor cells. It was proposed that platelets play a role in tumor progression and metastasis by reducing natural killer (NK) cell antitumor activity. Kopp et al. [46] showed that when coating tumor cells with platelet-derived soluble factors from stimulated platelets, the functions of NK cells were impaired. Additionally, the platelet coat might protect tumor cells from immunosurveillance [28]. In a study by Yavadav et al. [47], the GP6 signaling pathway was associated with endometrial cancer progression.

The other upregulated pathways included “regulation of the EMT in development pathway”, “actin cytoskeleton signaling”, “pulmonary fibrosis idiopathic signaling pathway”, and “integrin linked kinase (ILK) signaling”. Since Twist1 functions as an EMT inducer, it is a logical consequence that this pathway is upregulated. Actin filament modulation has also been closely associated with EMT [48] and the actin cytoskeleton has a vital role in completing EMT-induced alterations in the cells [49,50]. The transformations in the cell cytoskeleton are significant in several cancers. For example, changes to the actin cytoskeleton can promote metastasis [50]. Twist1 was shown to modulate the actin cytoskeleton in human glioblastoma [51]. Upregulation of the pulmonary fibrosis pathway seems reasonable since EMT also plays a role in pulmonary fibrosis [52]. ILK participates in many cell functions such as cell-extracellular matrix interactions, cell cycle, apoptosis, cell proliferation, and cell motility. ILK also has multiple functions in different cancers, such as inducing EMT [53,54]. Twist was proven to activate ILK, while in phyllodes breast tumors, ILK was shown to transmit its effects via the Twist pathway [35].

‘Hub genes’ are genes with a high degree of connectivity in the protein–protein interaction network that are significantly enriched in transcriptional regulation. From our differential expression analysis, we were able to highlight 28 genes that share a known protein to protein interaction. In “high Twist1”- samples, the transcriptionally downregulated molecules included the B-cell lineage markers *PAX5*, *CD19*, *CD22*, *CD20 (MS4A1) CD79a*, B-cell activator cytokine *TNFSF13B,* and the antigen presentation marker *CD40*, whereas transcriptionally upregulated molecules included cell–cell interaction molecules tight junction protein 1 (*TJP1*) and gap junction alpha-1 protein (*GJA1*), cell–matrix interaction molecule integrin alpha 1 (*ITGA1*) with multiple extracellular matrix proteins collagen type V alpha 2 (*COL5A2*), decorin (*DCN*), fibrillin (*FBN1*), transgelin (*TAGLN*), basement membrane protein laminins (*LAMA*, *LAMB*, *LAMC*), and extracellular matrix cross linker lysyl oxidase (*LOX).* Very few studies are available about cell interaction and matrix molecules in mycosis fungoides [55,56,57].

The observed downregulation of B-cell markers contrasts with previous papers reporting a trend towards worse prognosis in the presence of over 50% of CD20 positive cells [58] and upregulation of the CD20 gene (*MS4A1*) in MF disease progression [59]. The downregulation of B-cell markers could result from a paucity of reactive intratumoral B-lymphocytes or true gene expression downregulation in such cells.

One of the most interesting, downregulated pathways was the “Th 1 pathway”. During MF progression, the amount of Th1 cells decreases while the amount of Th2 cells increases [23,60,61]. This change in the predominance between Th1 and Th2 cells also changes the cytokine milieu of the tumor. Malignant T-cells produce immunoregulatory cytokines that repress Th1 responses and activate signaling pathways related to altered immune responses in the tumor microenvironment, further enhancing disease progression [17]. Based on hub gene analysis, one of the most strongly upregulated genes in the Twist1 high group is CD73 (*NT5E*), which is an integral protein in immune suppression [62]. The expression of CD22, a molecule that prevents autoimmune reactions, was reported to be expressed in MF [63] but our results were different.

Surprisingly, the extracellular-signal-regulated kinase (ERK)/MAPK signaling pathway was downregulated when Twist1 expression was high. Hyperactivation of this signaling pathway was detected in cancer development and progression [64]. Previously, the upregulation of MAPKs was associated with Twist1 overexpression in breast cancer [65] and melanoma [66]. The hub gene analysis highlighted the upregulation of gap-junction protein (GJA1), which is a limiting factor in MAPK/ERK signaling [55]. In MF, malignant cells form gap junctions with Langerhans cells [67].

Other downregulated pathways included “phosphoinositide 3–kinase (PI3K) signaling in B-lymphocytes”, “ribosomal protein S6 kinase beta 1 (p70S6K) signaling”, and “triggering receptor expressed on myeloid cells 1 (TREM1) signaling systemic lupus erythematosus in B-cell-signaling pathway”. The PI3K–Akt pathway is an intracellular signal transduction pathway that promotes metabolism, proliferation, cell survival, growth, and angiogenesis in response to extracellular signals. The PI3K/Akt signaling pathway has a connection with EMT, having the ability to influence tumor aggressiveness by affecting EMT [68]. Indeed, the hub gene analysis highlighted differentially expressed EMT related adhesion and matrix proteins, which is no surprise as the DE analysis was set against high and low expression of EMT transcription factor Twist1, and additionally, they are essential to the microdissected area.

## 5. Conclusions

In conclusion, Twist1 overexpression seems to be associated with several proteins and pathways involved in immunoregulation and lymphocyte differentiation. Zeb1 protein expression did not show any major correlation with global RNA expression in the PCA. However, there were only four cases with strong Zeb1 protein expression, a fact that precludes drawing any firm conclusions from these data.

One limitation of our study is that we used both diagnostic and follow-up samples. However, these two groups did not considerably differ in terms of their gene expression. In addition, the results of the DE analysis with all samples combined correlated well with those from the analysis using only diagnostic samples, thus indicating that this limitation likely did not have a major impact on the results. Additionally, the number of cases was limited; especially regarding the analyses of Zeb1 expression, limited number of cases may have hindered the detection of some existing biological differences. Additionally, using paraffin-embedded tissue may have interfered with the sensitivity of the method. The strength of our study is in the high standard data analyses as well as the use of laser-captured microdissected samples, which decreased the bias caused by non-malignant stromal and epithelial cells. However, we were not able to fully rule out the impact of dilution of genes of interest. Could there, with this approach, be a dilution of genes of interest in early-stage disease as the microdissected area represents a much larger percentage of stromal and non-neoplastic T-cells compared to MF in advanced-stage disease? We find this unlikely, since Twist1 expression did not correlate with MF staging or tumor cell density or percentage of reactive lymphocyte. Although we microdissected CD3–positive cells from the FFPE sections, cell populations with different backgrounds were inevitably collected during RNA extraction.

Considering the present results compared to recent literature, we anticipate Twist1 to be a central transcription factor and pathway regulator in the disease progression of MF. Naturally, in this kind of experiment setting, we were not able to confirm causality, and our results still need to be validated in cell culture or animal models with Twist1 knockout. Nevertheless, these results suggest that Twist1 is an interesting object for developing targeted therapies for MF.

## Figures and Tables

**Figure 1 cancers-15-01527-f001:**
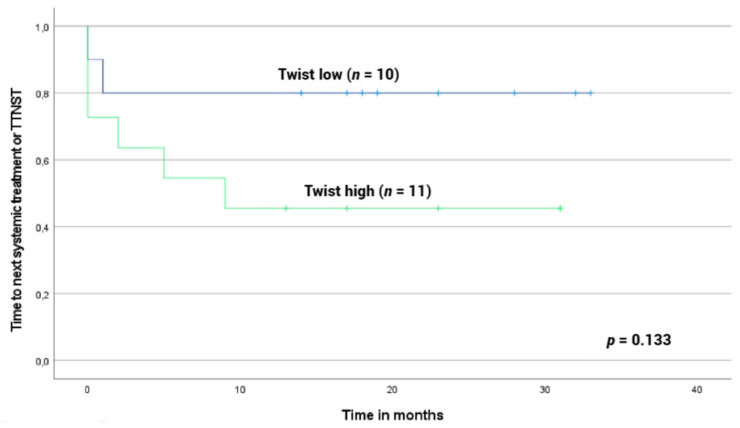
Time to systemic therapy (TTST) based on Twist1 nuclear expression in MF patient samples. TTST was evaluated from confirmed diagnosis date to beginning of systemic therapy or last follow-up date. The cut-off value for Twist1 was 17.6% and cases were divided into two groups based on expression results: Twist1 low <17.6% (*n* = 10) and Twist1 high ≥17.6% (*n* = 11). Patients with high nuclear Twist1 expression were associated with shorter TTST and required systemic treatments earlier (*p* value = 0.133).

**Figure 2 cancers-15-01527-f002:**
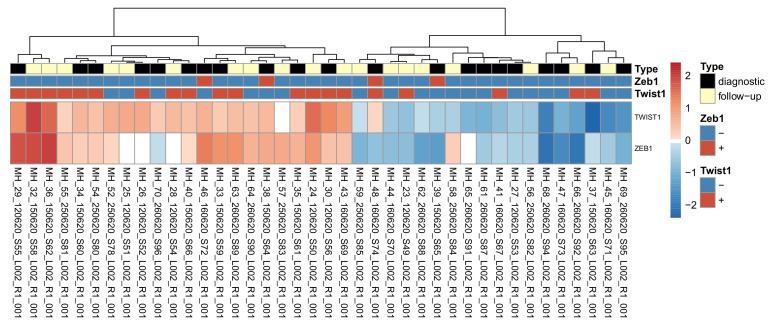
Comparison of immunohistochemical and RNA levels of Twist1 and Zeb1 in analyzed samples. First three rows indicate diagnostic and follow-up samples, Twist+ and Twist- and Zeb+ and Zeb- immunohistochemistry. Heat map of normalized read counts (normalized with the VST method in DESeq2) for all analyzed samples for *TWIST1* and *ZEB1* RNA. The rows are scaled so that blue indicates below-average expression for the gene, and red indicates above-average expression.

**Figure 3 cancers-15-01527-f003:**
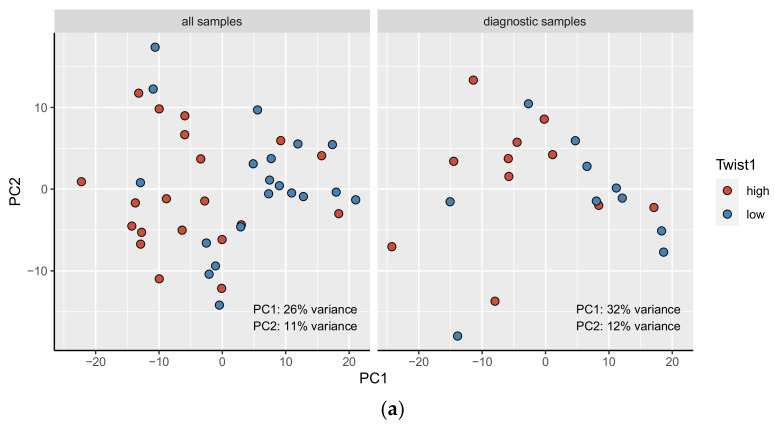
First two principal components of the PCA, with Twist1 (**a**) and Zeb1 (**b**) expression categories for all samples (*n* = 40) and diagnostic samples only (*n* = 21). Blue indicates low Twist1/Zeb1 expression, and red indicates high Twist1/Zeb1 expression.

**Figure 4 cancers-15-01527-f004:**
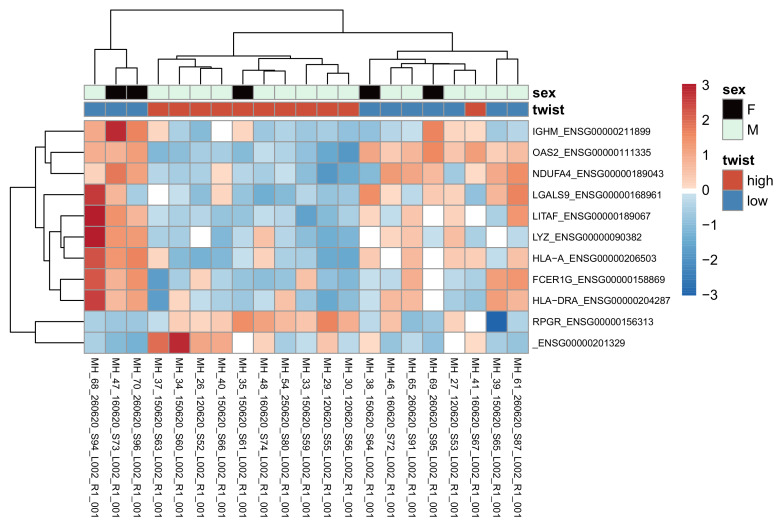
Heatmap of normalized read counts (with the VST method in DESeq2) for the nine genes according to adjusted *p* value, for diagnostic samples in the DE analysis. The rows are scaled so that blue indicates below-average expression for the gene, and red indicates above-average expression. IGHM, Immunoglobulin heavy constant; Mu OAS2, 2′-5′-oligoadenylate synthetase; NDUFA4, NDUFA4 Mitochondrial complex associated; LGALS9, Galectin 9; LITAF, Lipopolysaccharide induced TNF factor; LYZ, Lysozyme; HLA-A, Major histocompatibility complex, class I; FCER1G, Fc epsilon receptor Ig; HLA-DRA, Major histocompatibility complex, class II, DR Alpha; RPGR, Retinitis pigmentosa GTPase regulator.

**Figure 5 cancers-15-01527-f005:**
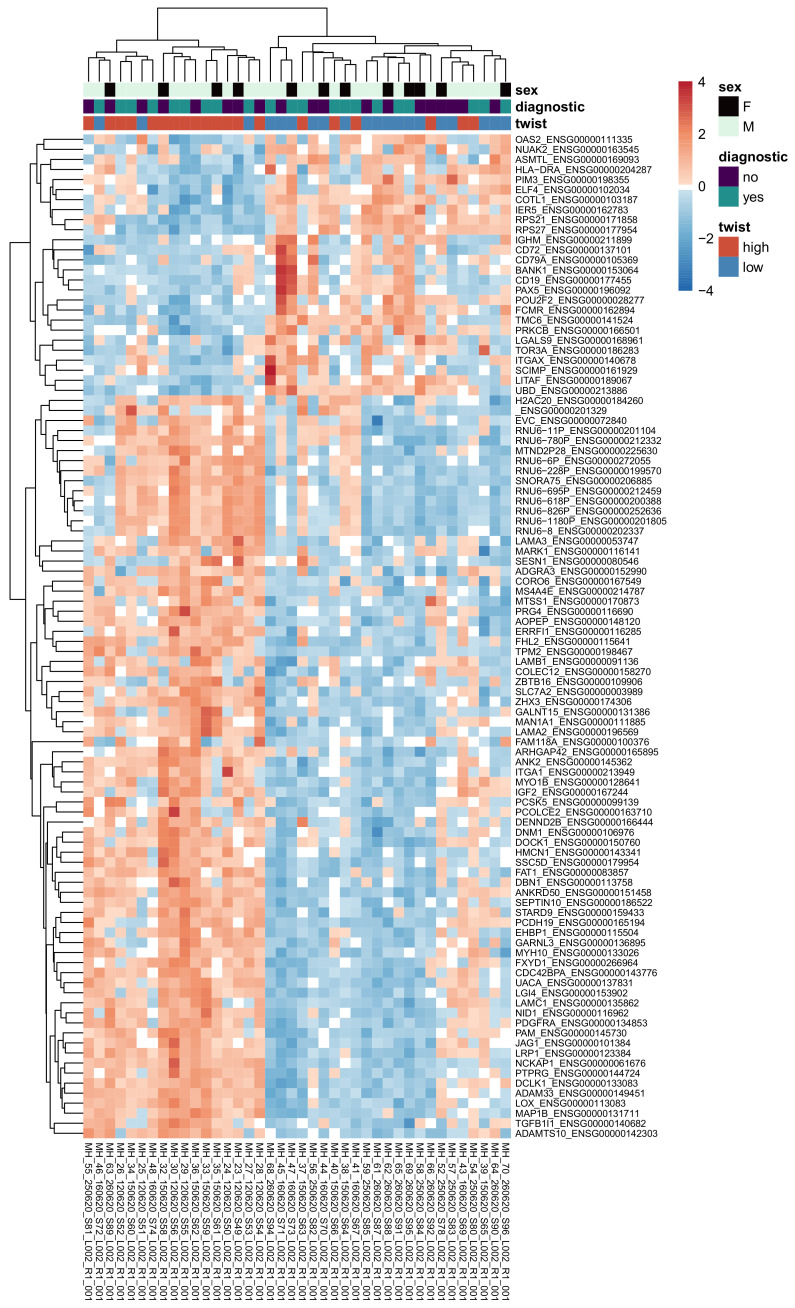
Heatmap of normalized read counts (with the VST method in DESeq2) for the top 100 genes according to adjusted *p* value, for all samples in the differential expression analysis. The rows are scaled so that blue indicates below-average expression for the gene, and red indicates above-average expression.

**Figure 6 cancers-15-01527-f006:**
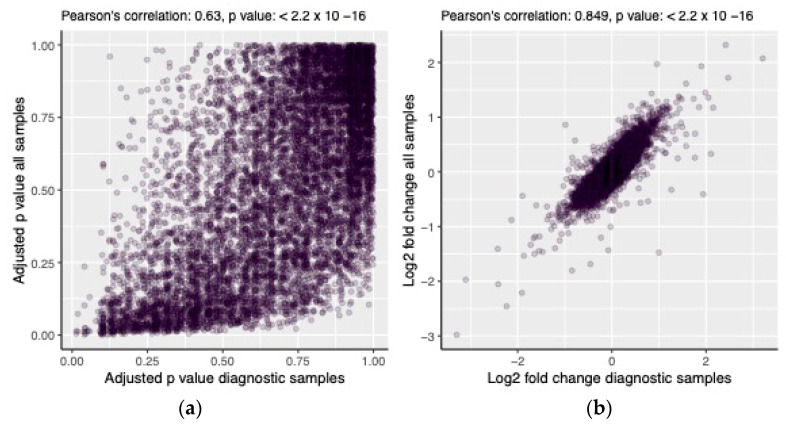
Comparison of the results of the DE analysis run on diagnostic samples and all samples combined, showing the relationship between the two models’ adjusted *p* values (**a**) and predicted log2 fold changes (**b**).

**Figure 7 cancers-15-01527-f007:**
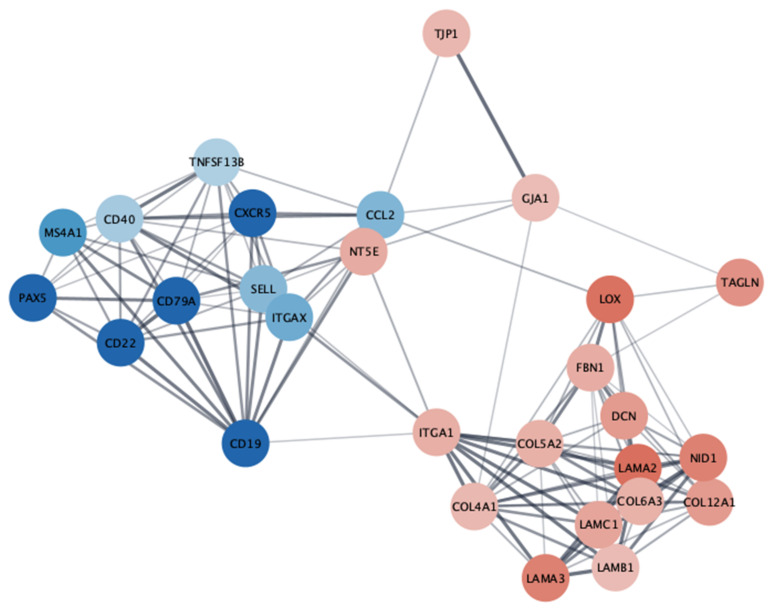
Interaction network of the 28 identified hub genes. The fill color corresponds to log2 fold change estimated by DESeq2, with blue indicating negative changes and red indicating positive changes in the high-Twist1-expression group relative to the low-expression group (created using the Cytoscape software). *CCL2*, C–C motif chemokine ligand2; *CD*, cluster of differentiation; *COL5A2*, collagen type V alpha 2 chain; *CXCR5*, C–X–C chemokine receptor type 5; *DCN*, decorin; *FBN1*, fibrillin 1; *GJA1*, gap junction alpha–1 protein; *ITGAX*, integrin subunit alpha X; *ITGA1*, integrin subunit alpha 1; *LAMA2*, laminin subunit alpha 2; *LAMA3*, laminin subunit alpha 3; *LAMB1*, laminin subunit beta 2; *LAMC1*, laminin subunit gamma 1; *LOX*, lysyl oxidase; *MS4A1*, membrane spanning 4-domains A1; *NT5E,* 5′-nucleotidase ecto; *PAX5*, paired box 5; *SELL*, selectin L; *TAGLN*, transgelin; *TJP1*, tight junction protein 1; *TNFSF13B*; TNF superfamily member 13b, *NID*, Nidogen 1.

**Table 1 cancers-15-01527-t001:** Patient demographics: The median follow-up time was 32.2 months (range: 6.28–203 months). 6 duplicated patient samples were removed.

	Diagnostic Samples, *n* (%)	Follow-up Samples, *n* (%)	All Samples, *n* (%)
**Number of cases**	21	19	40
**Male**	16/21 (76%)	12/19 (63%)	28/40 (70%)
**Age 60 years or older**	16/21 (76%)	11/19 (58%)	27/40 (68%) (median 63 years, range 19–86)
**Stage I–IIA**	16/21 (76%)	13/19 (68%)	29/40 (73%)
**Stage IIB–IV**	5/21 (24%)	6/19 (32%)	11/40 (28%)
**Elevated LDH**	7/21 (33%)	7/19 (37%)	14/40 (35%)
**Presenting lesions**			
**Solitary**	1/21 (5%)	0/19 (0%)	1/40 (3%)
**Multiple**	16/21 (76%)	15/19 (79%)	31/40 (78%)
**Erythrodermic**	4/21 (19%)	4/19 (21%)	8/40 (20%)

LDH, Lactate dehydrogenase.

## Data Availability

For detailed data, please contact corresponding author.

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
