# Peer review of "The Transcription Factor Twist1 Has a Significant Role in Mycosis Fungoides (MF) Cell Biology: An RNA Sequencing Study of 40 MF Cases"

_cancers, 2023, doi:10.3390/cancers15051527_

Round 1

Reviewer 1 Report

Comments to the Authors:

Summary: The manuscript entitled “The Transcription Factor Twist1 Has a Significant Role in Mycosis Fungoides (MF) Cell Biology: An RNA Sequencing Study of 40 MF Cases” by Häyrinen MJ et al. aims to investigate the prognostic value of the transcription factors Twist1 and Zeb1 in mycosis fungoides (MF). RNA sequencing was performed on microdissected CD3 positive areas in skin biopsies from patients with different stages of MF. Additionally, Twist1 and Zeb1 protein expression were evaluated by immunohistochemistry and methylation level of the Twist1 promotor region were performed.

This study is an extension of previous findings from the same group, which were published as a letter to the editor, entitled “Twist and Zeb1 expression identify mycosis fungoides patients with low risk of disease progression”.

Although this paper contributes with important knowledge to the research field of MF, the paper lacks a more comprehensive description of the methods used, as well as a more clear presentation of the results.  

Major comments:

Material and Methods:

2.1. Patient Material:

·       Patient material consisted of 40 skin tumor biopsies from 40 MF patients with stage I-IV disease. What does the authors mean here? – Tumor biopsies from early stage disease? – Also described in line 91-92.

·       There were 21 diagnostic and 19 follow-up samples. Could the authors provide a more detailed information regarding the follow-up samples? – How many of the patients with stage I-IIA in the diagnostic sample progressed to a higher disease stage (IIB-IV)? How many had the “same stage” in both the diagnostic and follow-up sample. Were there any patients with lower disease-stage in the follow-up sample? Maybe this information could appear more clearly in Table 1?

 Immunohistochemical Staining, Analysis, And Correlation with Disease Presentation and Outcome:

·       The authors refer to their previous publication “Twist and Zeb1 expression identify mycosis fungoides with low risk of disease progression” about previously immunohistochemical staining procedure. In the referenced article, the immunohistochemical staining methods regarding Twist1 and Zeb1 are not described. The authors should provide a description of the immunohistochemical procedure performed including information about the specific antibodies used in this study.

·       How was the tumor cell count performed? – Manually or digitally? If manually – how many persons did the count? – Independently? Please provide a more detailed information for this procedure.

·       The authors describe that the tumor cell count was estimated as a percentage of the surrounding reactive lymphocytic infiltrate. Was this only done in the dermis? Did the authors also perform the tumor cell count in epidermis? - And how was the percentage of the “surrounding reactive lymphocytic infiltrate” done in epidermis? Moreover, in early-stage disease it can be quite difficult to distinguish reactive and neoplastic lymphocytes from each other.

Microdissection and RNA extraction:

·       Line 116: “In total 8-13 sequential paraffin embedded slide sections..” Why 8-13 slides? – is this range defined by the cell density in the different stages?

·       Did the authors apply the indicated microdissected “tumor surface area” of 5x106 µm2 for all slides? – was this area also applicable for early-stage disease samples? And on what basis was this tumor surface area defined?

The Twist1 Promotor methylation analysis:

·       After LCM, DNA was isolated. How many of the 8-13 slides were used for DNA isolation? And did the authors separate the microdissected tissue from the Eppendorf tube in two equal parts for RNA and DNA extraction?

Results:

3.2. Immunohistochemistry of Twist1 and Zeb1, Correlation with Histomorphology, Disease Presentation and Outcome:

·       Which cells were Twist1 and Zeb1 positive? - Was it only the neoplastic T-cells or also reactive lymphocytes? Which were counted as positive – all of them or only the morphologically assessed neoplastic cells?

Figure 3(a) and (b):

This figure is difficult to read and interpret. Could the authors provide a more readable figure were the definition and colors of patient samples, high/low Twist1 and Zeb1 expression are more separated and thus easier to read? 

Figure 7:

I suggest that this figure should be in the supplementary material.

Discussion:

Could the authors elaborate more of the impact/challenges of microdissection of CD3 positive areas in various MF disease stages? – Could there, with this approach, be a dilution of genes of interest in early-stage disease as the microdissected area represents a much larger percentage of stromal and non-neoplastic T-cells (as seen in case 32) compared to MF in advanced-stage disease?

Could the authors further discuss other aspects that could influence the results of Twist1 and Zeb1 expression in the different stages of MF?

Minor comments:

Line 27: “Additionally, high Twist1 immunohistochemical expression was…” could or should it be protein expression instead of immunohistochemical expression?

Line 117: “…and mounted on pet slides..” Please define “pet” first time used in the text.

Line 181: “…between Twist+ and Twist1-..”Should it be Twist1+?

Reviewer 2 Report

The presented manuscript uses RNA sequencing to investigate the biological mechanism underlying Twist1 and Zeb1 influence on mycosis fungoides (MF) prognosis. The authors show that Twist1 overexpression is associated with several proteins and pathways involved in immunoregulation and lymphocyte differentiation. Taking results of other authors together with results presented in this manuscript, the authors anticipate Twist1 to be a central transcription factor and pathway regulator in the disease progression of MF. Thus results presented in this study brings more insight into MF biology and suggest Twist1 as an interesting object for developing targeted therapies for MF.

The presented manuscript is well-written. The introduction provides sufficient background, the results are clearly presented as well as extensively and critically discussed. I would like to congratulate  the authors on their work.

Round 2

Reviewer 1 Report

Dear Authors

Thank you for the revised manuscript and coverletter. Your corrections are fully accepted for publication. Congratulations with this interesting paper.